# CARD8 SNP rs11672725 Identified as a Potential Genetic Variant for Adult-Onset Still’s Disease

**DOI:** 10.3390/life11050382

**Published:** 2021-04-23

**Authors:** Wei-Ting Hung, Yi-Ming Chen, Shuen-Iu Hung, Hsin-Hua Chen, Ning-Rong Gung, Chia-Wei Hsieh, Kuo-Tung Tang, Der-Yuan Chen

**Affiliations:** 1Institute of Clinical Medicine, National Yang Ming Chiao Tung University, Taipei 11221, Taiwan; wthung@vhgtc.gov.tw; 2Department of Medical Education, Taichung Veterans General Hospital, Taichung 40705, Taiwan; 3Department of Medical Research, Taichung Veterans General Hospital, Taichung 40705, Taiwan; ymchen1@vghtc.gov.tw (Y.-M.C.); shc5555@vghtc.gov.tw (H.-H.C.); 4School of Medicine, College of Medicine, National Yang Ming Chiao Tung University, Taipei 11221, Taiwan; 5Rong Hsing Research Center for Translational Medicine & Ph.D. Program in Translational Medicine, National Chung Hsing University, Taichung 40227, Taiwan; chiaweih@gmail.com (C.-W.H.); crashbug1982@gmail.com (K.-T.T.); 6Cancer Vaccine and Immune Cell Therapy Core Laboratory, Chang Gung Immunology Consortium, Chang Gung Memorial Hospital, Linkou, Taoyuan 33305, Taiwan; hungshueniu@gmail.com; 7Rheumatology and Immunology Center, Translational Medicine Laboratory, China Medical University Hospital, Taichung 40447, Taiwan; lotis0713@gmail.com; 8Division of Allergy, Immunology and Rheumatology, Taichung Veterans General Hospital, Taichung 40705, Taiwan; 9Rheumatology and Immunology Center, China Medical University Hospital, Taichung 40447, Taiwan; 10School of Medicine, China Medical University, Taichung 40447, Taiwan

**Keywords:** CARD8, single-nucleotide polymorphism, rs11672725, NLRP3-inflammasome signaling, adult-onset Still’s disease (AOSD)

## Abstract

Adult-onset Still’s disease (AOSD), an autoinflammatory disorder, is related to the dysregulation of NLR3-containing a pyrin domain (NLRP3)-inflammasome signaling. We aimed to investigate the associations of genetic polymorphisms of NLRP3-inflammasome signaling with AOSD susceptibility and outcome and to examine their functional property. Fifty-three candidate single-nucleotide polymorphisms (SNPs) involved in NLRP3-inflammasome response were genotyped using Sequenom MassArray on the samples from 66 AOSD patients and 128 healthy controls. The significant SNPs were validated by direct sequencing using a TaqMan SNP analyzer. Serum levels of associated gene products were examined by ELISA. One SNP rs11672725 of CARD8 gene was identified to be significantly associated with AOSD susceptibility by using MassArray and subsequent replication validation (*p* = 3.57 × 10^−7^; odds ratio 3.02). Functional assays showed that serum CARD8 levels were significantly lower in AOSD patients (median, 10,524.6 pg/mL) compared to controls (13,964.1 pg/mL, *p* = 0.005), while levels of caspase-1, IL-1β and IL-18 were significantly higher in patients (107.1 pg/mL, 2.1 pg/mL, and 1495.8 pg/mL, respectively) than those in controls (99.0 pg/mL, 1.0 pg/mL, and 141.4 pg/mL, respectively). Patients carrying rs11672725CC genotype had significantly higher serum caspase-1 and IL-18 levels (121.3 pg/mL and 1748.6 pg/mL) compared to those with CT/TT genotypes (72.6 pg/mL, *p* = 0.019 and 609.3 pg/mL, *p* = 0.046). A higher proportion of patients with rs11672725CC genotype had a systemic pattern of disease outcome, which was linked to low CARD8 levels. A novel variant, rs11672725, of the CARD8 gene was identified as a potential genetic risk for AOSD. Patients carrying the rs11672725CC genotype and C allele had low CARD8 levels, and were predisposed to a systemic pattern with an elevated expression of inflammasome signaling.

## 1. Introduction

Adult-onset Still’s disease (AOSD), a systemic inflammatory disorder, is characterized by spiking fever, rash, arthritis, lymphadenopathy, hepatosplenomegaly, variable multisystemic involvement, and increases in acute phase reactants [1,2]. AOSD has been considered an autoinflammatory disease due to its characteristic phenotypes without detectable autoantibodies [3,4]. Accumulating evidence indicates that NLR3-containing a pyrin domain (NLRP3)-inflammasome plays a pathogenic role in autoinflammatory diseases [5,6]. Moreover, a good response to interleukin (IL)-1β inhibitors and IL-18 binding protein observed in AOSD patients [7,8] suggests a critical role of NLRP3-inflammasome signaling in its pathogenesis. Our recent study also reveals an elevated expression of NLRP3-inflammasome signaling and their positive correlation with disease activity in AOSD suggest its involvement in the pathogenesis [9].

The pathogenesis of AOSD was considered to result from immune dysregulation, interaction between host and environment factors, and genetic complexity [10,11,12,13,14,15].

The importance of inflammasomes in regulating immune responses was recognized with the findings of polymorphisms in gene coding inflammasome signaling and their linkage to aberrant production of IL-1β and IL-18 in autoinflammatory diseases. Sequence variants in the NLRP3 gene are responsible for autoinflammatory diseases such as Muckle-Wells syndrome and familial cold autoinflammatory syndrome [16,17]. Over 40 mutations within *NLRP3* gene have been identified that are associated with autoinflammatory diseases, which result in the activation of NLRP3 inflammasome with production of bioactive IL-1β and IL-18 [18,19,20]. Although our studies and previous reports have shown a significant association of IL-18 promoter polymorphisms with the susceptibility of AOSD [14,15], the association of gene polymorphisms of NLRP3-inflammasome components with the susceptibility of AOSD has not been reported.

In this pilot study, we aimed to investigate: (1) an association of genetic polymorphisms of NLRP3-inflammasome signaling with AOSD susceptibility using a MassArray and subsequent replication; (2) the associations of genotypes and alleles with clinical manifestations and disease outcome; and (3) the functional association of the involved gene polymorphisms in AOSD.

## 2. Materials and Methods

### 2.1. Subjects

Sixty-six consecutive AOSD patients fulfilling the Yamaguchi criteria [21] were enrolled. Patients with infections, malignancies, or other rheumatic diseases were excluded. After investigation, all AOSD patients received treatment including corticosteroids and non-steroidal anti-inflammatory drugs (NSAIDs). The used disease-modifying anti-rheumatic drugs (DMARDs) were methotrexate, hydroxychloroquine, sulfasalazine, cyclosporine, and azathioprine. We obtained serum samples from AOSD patients were after treatment. According to the proposed classification of disease courses of AOSD [22,23], the AOSD patients, followed for at least one year, were classified into two patterns of disease course: a systemic pattern that includes the monocyclic and the polycyclic form, and another chronic articular pattern. Disease activity of each patient was assessed with Pouchot score, which assigned 1 point to each of 12 manifestations: fever, typical rash, pleuritis, pneumonia, pericarditis, hepatomegaly or abnormal liver function tests, splenomegaly, lymphadenopathy, leukocytosis > 15,000/mm^3^, sore throat, myalgia, and abdominal pain (maximum score: 12 points) [22]. One hundred twenty-eight ethnically and geographically matched healthy subjects who had no self-report of rheumatic diseases as the population controls from a biobank. This study was approved by the Institutional Review Board of our Hospital (approval number: CF11309), and each participant’s written consent was obtained according to the Declaration of Helsinki.

### 2.2. SNPs Selection and Genotyping Using MassArray

Genomic DNA was extracted from peripheral blood of the enrolled subjects using Genomic DNA Extraction kits (RBCbioscience, New Taipei City, Taiwan). We selected 53 SNPs from the candidate genes involved in NLRP3-inflammasome response (22 SNPs of *NLRP3,* 2 SNPs of Mediterranean fever (*MEFV*), 16 SNPs of *CARD8*, 3 SNPs of *caspase-1*, and 10 SNPs of *IL-1β*) based on the reported literature in the related inflammatory diseases [24,25,26,27,28,29] (Table 1). To select the most representative SNPs by capturing the majority of genetic variations, SNP genotype information was downloaded from the HapMap database (http://www.hapmap.ncbi.nlm.nih.gov/, accessed on 2 July 2014) and the National Center for Biotechnology Information dbSNP database (http://www.ncbi.nlm.nih.gov/snp, accessed on 2 July 2014). Tag SNPs were selected for *NLRP3*, *MEFV*, *CARD8*, *caspase-1*, and *IL-1β* using the criterion of minor allele frequency (MAF) >10%. The genotyping of all of 53 SNPs was performed by Sequenom MassArray system with matrix-assisted laser desorption inoisation-time-of-flight mass spectrometry (MALDI-TOF; San Diego, CA, USA) according to the manufacturer’s instructions. The mass spectrograms were analyzed using MassArray Typer software (Sequenom, San Diego, CA, USA). In each sample, we excluded SNPs with a call rate <95% or deviation from Hardy–Weinberg equilibrium (HWE) proportions (*p* < 0.01) in the control subjects.

### 2.3. Replication Analysis and Target Gene Sequencing

To further validate the findings from MassArray for 53 candidate SNPs, one statistically significant SNP was evaluated by direct sequencing using TaqMan SNP analyzer (Applied Biosystems, Foster city, CA, USA): rs11672725 SNP of CARD8 gene region (TaqMan assay: C__11708120_10). The PCR reaction was performed in the following steps: 10 min at 95 °C, then 40 cycles of 15 s at 95 °C, and followed by 1 min at 60 °C; the PCR buffer used TaqMan Universal Master Mix II, no UNG (Applied Biosystems, Foster City, CA, USA).

### 2.4. Determination of Serum Levels of Soluble CARD8, Caspase-1, and IL-1β and IL-18 by Using Enzyme-Linked ImmunoSorbent Assay (ELISA)

Serum levels of soluble CARD8, caspase-1, IL-1β, and IL-18 were determined by using ELISA kits (CARD8, MyBioSource, San Diego, CA, USA; Caspase-1, R&D Systems, Inc., Minneapolis, MN, USA; IL-1β, RayBiotech Inc., Norcross, GA, USA; IL-18, Medical & Biology Laboratories Co, Ltd., Naka-ku, Nagoya, Japan) respectively according to each of the manufacturer’s instructions.

### 2.5. Statistical Analysis

For the genotyping of 53 candidate genes, we conducted the statistical analysis for the associations by comparing the allele or genotype frequencies between AOSD cases and controls using the dominant-inheritance model, the recessive-inheritance model, or the additive model. The SNP associations were examined by Fisher’s exact tests and rank-ordered according to the lowest *p* value in these models. All of the *p* values were two-tailed. A *p*-value of less than 0.05 was considered to be statistically significant. The odds ratios (ORs) were calculated using Haldane’s modification [30]. The nonparametric Mann–Whitney U test was used for between-group comparison of expression levels of CARD8 and NLRP3-inflammasome components. The correlation coefficient was calculated using the nonparametric Spearman’s rank correlation test. The differences in the frequencies of significant alleles among AOSD patients with different patterns of disease course were examined using the Fisher’s exact test.

## 3. Results

### 3.1. Clinical Characteristics of AOSD Patients

Among 66 AOSD patients (age at study entry, mean ± standard deviation (SD), 36.5 ± 13.5 years; 46 women and 20 men), the presence of spiking fever (≥9 °C), evanescent rash, arthralgia, sore throat, arthritis, lymphadenopathy, and hepatosplenomegaly were observed in 60 (90.9%), 56 (84.8%),54 (81.8%), 53 (80.3%), 27 (40.9%), 25 (37.9%), and five (7.6%) respectively. However, there were no significant differences in demographic data between AOSD patients and healthy controls (36.1 ± 9.1 years; 97 women and 31 men).

### 3.2. Differentially Expressed SNPs Using MassArray Analysis

All of 53 studied SNPs in AOSD cases and controls were in Hardy-Weinberg equilibrium displaying *p* > 0.05. Among the 53 SNPs of NLRP3-inflammasome signaling, the initial MassArray discovered only SNP rs11672725 of CARD8 gene region was significantly associated with the susceptibility of AOSD patients (Table 1). 

### 3.3. Replication Analysis and Target-Sequencing of the Candidate Gene

We replicated the association of the differentially expressed SNP rs11672725 of CARD8 gene found in the MassArray by direct sequencing in 66 AOSD cases and 128 healthy controls. Our results still showed that a significantly higher frequency of rs11672725CC genotype was found in AOSD patients compared to healthy controls (80.3% vs. 64.1%, *p* = 0.030). We also revealed a significant difference of AOSD susceptibility with CC genotype when compared to CT/TT genotype (Table 2).

Taking rs11672725 CC genotype as the reference, we observed a decrease of odds ratio (OR) of CT/TT genotype, in AOSD patients compared with healthy controls (OR, 95th CI: 0.44 (0.27–0.72), *p* = 0.001).

### 3.4. Association of SNP rs11672725 of CARD8 Gene with Serum Levels of CARD8, Caspase-1, IL-1β, and IL-18

As CARD8 gene encodes CARD protein, which is a negative regulator of NLRP3-inflammasome signaling [30], we examined the functional association of SNP rs11672725 by determining serum levels of CARD8 and inflammasome signaling molecules, including caspase-1, IL-1β, and IL-18. Our results showed significantly lower levels of CARD8 in AOSD patients (median, 10524.6 pg/mL, inter-quartile range (IQR): 5644.4-21,173.6 pg/mL) compared with the controls (median, 13,964.1 pg/mL, IQR: 11,219.6-20,051.3 pg/mL, *p* = 0.005), while higher levels of IL-1β, and IL-18 in AOSD patients (median, 2.1 pg/mL, IQR: 1.0–4.8 pg/mL; and 1495.8 pg/mL, IQR: 466.7–10,970.7 pg/mL; respectively) compared with the controls (median, 1.0 pg/mL, IQR: 0–1.7 pg/mL; and 141.4 pg/mL, IQR: 103.1–181.9 pg/mL; respectively, both *p* < 0.001). However, there was no significance in serum caspase-1 levels between AOSD patients and healthy controls (Figure 1).

Among AOSD patients, those carrying the rs11672725 CC genotype had a trend of lower levels of serum CARD8 (median, 9900.0 pg/mL, IQR: 5644.4–19,719.2 pg/mL) than those with CT/TT genotype (median, 19,202.2 pg/mL, IQR: 5404.9–26,324.9 pg/mL, *p* = 0.346) (Figure 2A). On the contrast, significantly higher levels of caspase-1 and IL-18 were noticed in AOSD patients carrying the CC genotype (median, 121.3 pg/mL, IQR: 72.6–251.7 pg/mL; and 1748.6 pg/mL, IQR: 492.6–12,433.9 pg/mL; respectively) compared with the carriers of CT/TT genotype (median, 72.6 pg/mL, IQR: 42.0–108.7 pg/mL, *p* = 0.019; and median, 609.3 pg/mL, IQR: 392.8–1550.6 pg/mL, *p* = 0.046; respectively) (Figure 2B,D). However, there was no significant difference in serum IL-1β levels between AOSD patients carrying CC genotype and CT/TT genotype.

### 3.5. Association of CARD8 SNP rs11672725 with Clinical Manifestations of AOSD 

As illustrated in Table 3, a significant higher proportion of AOSD patients carrying *CARD8* SNP rs11672725CC genotype had evanescent rash compared to those with CT/TT genotype respectively (90.6% versus. 61.5%, *p* = 0.02). However, there was no significant difference in the proportion of the other manifestations or clinical data between AOSD patients carrying CC genotype and CT/TT genotype.

### 3.6. Association of CARD8 SNP rs11672725 with Disease Outcome in AOSD Patients

Regarding the disease outcome, patients carrying the rs11672725 CC genotype had a significant higher proportion of systemic pattern of disease outcome compared with those carrying CT/TT genotype (83.0% versus 30.8%, *p* = 0.001, Table 3). 

To identify independent factors associated with systemic pattern of disease outcome, logistic regression analysis was performed (Table 4). Both in the univariate regression and multivariate forward stepwise model, serum levels of caspase-1 (OR: 1.01, 95% CI: 1.003–1.026, *p* = 0.011 and OR: 1.01, 95% CI: 1.002–1.025, *p* = 0.019, respectively) were significantly associated with a systemic pattern. Moreover, the rs11672725 CC genotype was an independent risk factor for the occurrence of a systemic pattern of disease outcome (OR: 11.11, 95% CI: 2.78–50.00, *p* = 0.001 and OR: 7.69, 95% CI: 1.67–33.33, *p* = 0.009).

## 4. Discussion

This study is the first to investigate the association of NLRP3-inflammasome gene polymorphisms with the susceptibility, clinical manifestations, and disease outcome of AOSD, which is a rare disease and has been considered as an autoinflammatory disease (3–4). We identified a novel genetic variant, the SNP rs11672725 of CARD8 gene, as a significant genetic variant for AOSD susceptibility by using both MassArray analysis and direct sequencing. The results showed significantly higher frequencies of SNP rs11672725CC genotype and C-allele in AOSD patients compared with healthy controls. Given that CARD8 is a negative regulator of NLRP3-inflammasome signaling [31], our functional analysis revealed lower levels of serum CARD8, while significant higher levels of caspase-1 and IL-18 in AOSD patients with CC genotype than those with CT/TT genotype. In logistic regression model, CC genotype could predict systemic pattern of disease outcome in AOSD patients. Based on these observations, the SNP rs11672725 of CARD8 gene may be a genetic variant linked with the susceptibility and adverse outcome of AOSD.

CARD8 (also known as TUCAN, tumor-up-regulated CARD-containing antagonist of caspase nine) is an important component of NLRP3-inflammasome, which regulates the production of downstream cytokines including IL-1β and IL-18 [30]. There is evidence to suggest that CARD8 may be a negative regulator of NF-kB. Mathews et al. demonstrated that rheumatoid arthritis patients carrying rs11672725T allele of CARD8 gene had higher baseline CARD8 protein levels compared to those without carrying this minor allele [25]. *CARD8* rs11672725TT genotype showed significant associations with H. pylori infection related gastric cancer [32]. In the present study, our results showed lower levels of serum CARD8 in AOSD patients carrying rs11672725CC genotype or C-allele compared with those carrying CT/TT genotype or T-allele. Based on these observations, we speculated that AOSD patients carrying rs11672725C allele or CC genotype of CARD8 gene might have an insufficient CARD8 levels to protect against the activation of NLRP3-inflammasome signaling. Our AOSD patients carrying SNP rs11672725CC genotype also had higher levels of IL-18 and capase-1 compared to those CT/TT genotype, supporting this hypothesis.

Despite new biomarker such as IL-1β and IL-18 are emerged in recent years [33], only few evidences demonstrate an association of biomarkers with manifestations of AOSD [34,35]. Our results showed AOSD patients carrying SNP rs11672725CC genotype have a significantly higher proportion of evanescent rash, which is a common manifestation of systemic pattern, when compared to those with SNP rs11672725CT/TT genotype.

In addition, the disease courses and prognosis of AOSD patients also vary considerably [35], and none of the clinical or laboratory variables could predict disease outcome. Previous studies reported higher levels of serum IL-18 was associated with systemic pattern in AOSD patients [15,36]. In the present study, a significantly higher proportion of AOSD patients carrying SNP rs11672725CC genotype had systemic pattern, which was associated with significantly higher levels of serum IL-18. Moreover, our results also showed positive correlation of serum caspase-1 level and systemic pattern in AOSD patients, which had not been reported in literature. 

There were some limitations in this study. We adjusted the *p*-value of 53 candidate genes in the discovery dataset with Benjamini–Hochberg correction. Neither *p*-value of CARS8 rs11672525 genotype or allele type were statistically significant (*p* = 1.0 and *p* = 0.477, respectively). Type I error was a concern and the sample size may be too small to draw a definitive conclusion regarding the associations of the SNP rs11672725 of CARD8 gene with clinical manifestations or disease outcome. However, it might not be readily achievable because AOSD is a rare disease: its prevalence has been estimated to be lower than one case per 100,000 people [37], placing it among the orphan diseases. We obtained serum samples after immunosuppressive treatment, and we cannot exclude the potential interferences of immunosuppressants to inflammatory cytokines (IL-1 and IL-18) or inflammatory markers (CRP and ferritin). Future studies of treatment-naïve AOSD patients are needed to fully address this issue. 

In this study, we did no demonstrate the influence of IL-18 binding protein (IL-18 BP) on IL-18 activity and levels of free IL-18 of serum from AOSD patients. Imbalance of IL-18/IL-18BP in patients was reported in several diseases, such as hemophagocytic syndrome [38] and systemic lupus erythematosus [39], but IL-18BP was only moderately elevated even under the active stage. It suggested that IL-18BP protein may be reactive to serum IL-18 level, high disease activity, and may play a balancing role in inflammatory diseases. Further study to elucidate the relation between IL-18 BP and AOSD may be needed.

We also did not enroll other ethnic groups or evaluate the impact of environmental factors that may play an important role in transfusing the genetic phenotypes into disease phenotypes. Future studies should enroll large cohorts of AOSD patients of different ethnicities and proceed with fine-mapping and functional studies of the other identified genetic variants.

## 5. Conclusions

We have identified a novel SNP rs11672725 of CARD8 gene to be associated with the susceptibility of Chinese patients with AOSD. Moreover, this SNP might be related to the occurrence of skin rash and disease outcome of AOSD. These results implicate that the genetic polymorphisms of SNP rs11672725 of CARD8 gene may be involved in AOSD aetiopathogenesis.

## Figures and Tables

**Figure 1 life-11-00382-f001:**
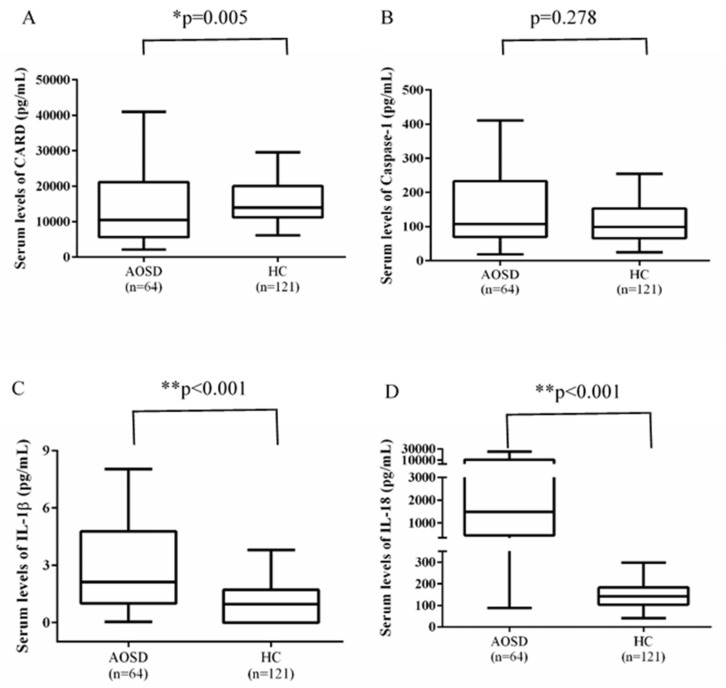
Functional associations of the genotypes or alleles of the rs11672725 of CARD8 gene with serum levels of NLRP3-inflammasome signaling molecules in patients with adult-onset Still’s disease (AOSD) and healthy controls (HC). The differences of serum levels of CARD8 (**A**), caspase-1 (**B**), IL-1β (**C**), and IL-18 (**D**). Data are presented as box-plot diagrams, with the box encompassing the 25th percentile (lower bar) to the 75th percentile (upper bar). The horizontal line within the box indicates median value respectively for each group. * *p* < 0.01, ** *p* < 0.001, versus HC.

**Figure 2 life-11-00382-f002:**
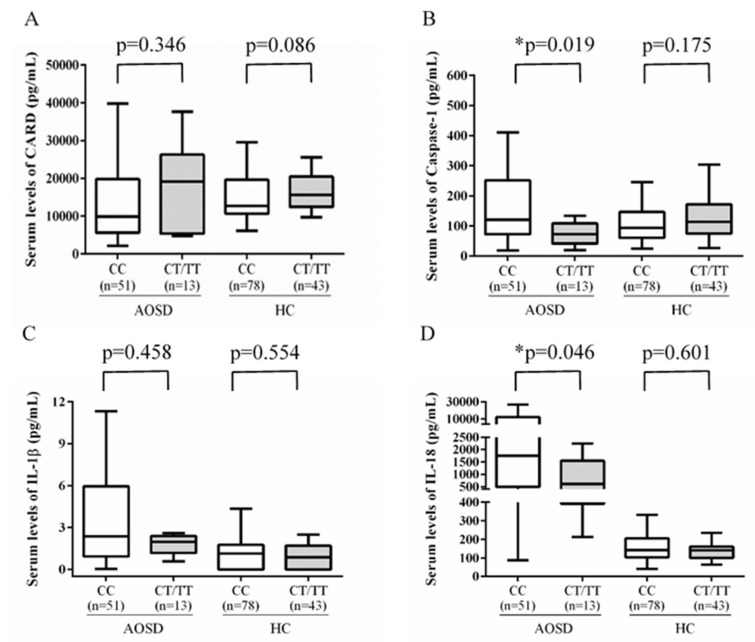
Serum level of CARD8 (**A**), caspase-1 (**B**), IL-1β (**C**), and IL-18 (**D**) between AOSD patients with the rs11672725CC genotype and those with the rs11672725CT/TT genotypes, or between healthy controls with the rs11672725CC genotype and those with the rs11672725CT/TT genotypes. Data are presented as box-plot diagrams, with the box encompassing the 25th percentile (lower bar) to the 75th percentile (upper bar). The horizontal line within the box indicates median value respectively for each group. * *p* < 0.05.

**Table 1 life-11-00382-t001:** The allele frequency of inflammasome genes in patients with adult-onset Still’s disease and health control.

Gene	Chromosome	SNP ID	Function	Minor	AOSD Case	Control	Genotype	Allele	HWE
	Position			allele	MAF	MAF	*p*-Value	*p*-Value	*p*-Value
*NLRP3*	1:247421341	rs10754555	intron	G	0.386	0.414	0.884	0.647	0.479
*NLRP3*	1:247435930	rs10754557	intron	A	0.179	0.129	0.502	0.5330	0.410
*NLRP3*	1:247432548	rs10925019	intron	T	0.276	0.320	0.687	0.467	0.244
*NLRP3*	1:247440956	rs10925026	intron	C	0.388	0.386	0.951	1	0.277
*NLRP3*	1:247417907	rs12137901	intron variant, upstream variant 2 KB	C	0.302	0.332	0.725	0.632	0.215
*NLRP3*	1:247446026	rs12565738	intron	T	0.086	0.051	0.189	0.244	0.219
*NLRP3*	1:247424175	rs3806268	synonymous codon	A	0.560	0.520	0.424	0.502	0.209
*NLRP3*	1:247417266	rs4925648	intron variant, upstream variant 2 KB	T	0.224	0.273	0.618	0.371	0.800
*NLRP3*	1:247420773	rs4925650	intron	A	0.466	0.441	0.787	0.736	0.158
*NLRP3*	1:247432755	rs4925654	intron	A	0.181	0.180	0.819	1	0.060
*near NLRP3*	1:247457731	rs6672995	NA (decrease IL1-β production)	A	0.060	0.039	0.412	0.423	0.646
*NLRP3*	1:247419919	rs7512998	intron	C	0.086	0.078	0.831	0.838	0.338
*NLRP3*	1:247443750	rs10159239	intron	G	0.397	0.445	0.453	0.429	0.394
*NLRP3*	1:247448734	rs10754558	utr variant 3 prime (3′UTR)	G	0.405	0.410	0.707	1	0.575
*NLRP3*	1:247420375	rs10925015	intron	C	0.466	0.438	0.680	0.653	0.106
*NLRP3*	1:247425329	rs121908148	missense/Pathogenic (germline)	G	0.009	0.000	0.312	0.312	1
*NLRP3*	1:247438293	rs12239046	intron	T	0.388	0.387	0.926	1	0.242
*NLRP3*	1:247424507	rs28937896	missense/Pathogenic (germline)	C	0.000	0.000	1	1	1
*NLRP3*	1:247425556	rs35829419	missense	A	0.000	0.000	1	1	1
*NLRP3*	1:247423034	rs3806265	intron	C	0.440	0.480	0.424	0.502	0.209
*NLRP3*	1:247435768	rs4612666	intron	T	0.414	0.473	0.403	0.313	0.572
*NLRP3*	1:247440161	rs4925659	intron	A	0.534	0.523	0.977	0.911	0.463
*MEFV*	16:3243888	rs1231122	missense, stop gained, synonymous codon	T	0.362	0.355	0.891	0.908	0.221
*MEFV*	16:3246429	rs224204	intron	T	0.612	0.613	0.926	1	0.242
*CARD8*	19:48253518	rs10403848	intron	A	0.544	0.316	0.142	1	0.940
*CARD8*	19:48239130	rs10416565	intron	G	0.026	0.031	1	1	0.715
*CARD8*	19:48219568	rs10418239	intron	G	0.233	0.281	0.676	0.376	0.412
*CARD8*	19:48218923	rs10500299	intron variant, nc transcript variant, synonymous codon	T	0.216	0.137	0.064	0.068	0.769
*CARD8*	19:48243424	rs11672725	intron variant, upstream variant 2 KB	T	0.079	0.209	0.024	0.009	0.296
*CARD8*	19:48231685	rs11881179	reference	C	0.000	0.000	1	1	1
*CARD8*	19:48251529	rs12460147	intron variant, upstream variant 2 KB	A	0.000	0.000	1	1	1
*CARD8*	19:48232193	rs16981832	intron	A	0.052	0.035	0.757	0.761	0.680
*CARD8*	19:48234765	rs16981845	intron	C	0.080	0.037	1	1	0.674
*CARD8*	19:48241088	rs16981853	intron variant, upstream variant 2 KB	C	0.704	0.541	0.065	0.250	0.572
*CARD8*	19:48249263	rs16981864	intron	G	0.355	0.282	0.226	0.174	0.405
*CARD8*	19:48255548	rs17589988	intron variant, upstream variant 2 KB	G	0.000	0.000	1	1	1
*CARD8*	19:48234294	rs2288877	intron	G	0.224	0.156	0.080	0.142	0.451
*CARD8*	19:48234449	rs2043211	missense, nc transcript variant, stop gained, utr variant 5 prime (5′UTR)	T	0.517	0.543	0.898	0.655	0.420
*CARD8*	19:48240320	rs6509366	intron	A	0.800	0.535	0.897	0.823	0.634
*CARD8*	19:48233449	rs4802449	intron	A	0.462	0.188	0.131	0.131	0.772
*CASP1*	11:105029658	rs501192	intron	A	0.009	0.016	1	1	0.858
*CASP1*	11:105032992	rs572687	intron	T	0.000	0.012	0.553	0.551	0.893
*CASP1*	11:105035212	rs3809024	intron variant, upstream variant 2 KB	A	0.000	0.000	1	1	1
*IL-1β*	2:112838252	rs1143623	upstream variant 2 KB	G	0.595	0.609	0.955	0.819	0.359
*IL-1β*	2:112836810	rs1143627	upstream variant 2 KB/Pathogenic	T	0.543	0.559	0.862	0.822	0.146
*IL-1β*	2:112832890	rs1143633	intron	A	0.578	0.571	0.412	0.910	0.559
*IL-1β*	2:112832813	rs1143634	synonymous codon	T	0.017	0.012	0.6480	0.649	0.893
*IL-1β*	2:112830976	rs1143642	intron	T	0.000	0.000	1	1	1
*IL-1β*	2:112830725	rs1143643	intron	A	0.552	0.512	0.771	0.503	0.380
*IL-1β*	2:112837290	rs16944	upstream variant 2 KB	G	0.544	0.563	0.957	0.735	0.148
*IL-1β*	2:112833698	rs3136558	intron	C	0.362	0.359	0.445	1	0.857
*IL-1β*	2:112834786	rs3917356	intron	A	0.466	0.484	0.730	0.823	0.160
*near IL-1β*	2:112840530	rs4848306	NA	A	0.500	0.508	0.881	0.911	0.29

MAF: minor allele frequency; UTR: untranslated region; HWE: Hardy-Weinberg equilibrium NC: non-coding RNA. *p*-value, versus healthy controls, was determined by Fisher Exact Probability Test (genotype), Fisher’s Exact Test (allele) and HWE.

**Table 2 life-11-00382-t002:** The association of CARD8 SNP rs11672725 CC-genotype and C-allele with susceptibility of adult-onset Still’s disease.

rs11672725	AOSD (*n* = 66)		HC (*n* = 128)		*p* Value	OR	95% CI	*p* Value ^#^
Genotype					0.066			
CC	53	(80.3%)	82	(64.1%)		ref.		
CT	11	(16.7%)	38	(29.7%)		0.45	(0.26–0.76)	0.003 **
TT	2	(3.0%)	8	(6.3%)		0.39	(0.13–1.19)	0.097
Genotype					0.030 *			
CC	53	(80.3%)	82	(64.1%)		ref.		
CT + TT	13	(19.7%)	46	(35.9%)		0.44	(0.27–0.72)	0.001 **
Allele type					0.025 *			
C	117	(88.6%)	202	(78.9%)		ref.		
T	15	(11.4%)	54	(21.1%)		0.48	(0.26–0.89)	0.019 *

* *p* < 0.05, ** *p* < 0.005, versus HC, was determined by Chi-Square test and # Logistic regression.

**Table 3 life-11-00382-t003:** The association of *CARD8* SNP rs11672725 CC-genotype with clinical manifestations and disease outcome in patients with AOSD.

	CC (n = 53)		CT/TT (n = 13)		*p* Value
Age	32.0	(23.5–45.5)	37.0	(28–48.5)	0.239
Female	36	(67.9%)	12	(92.3%)	0.094
Rash ^f^	48	(90.6%)	8	(61.5%)	0.020 **
Arthralgia ^f^	44	(83.0%)	11	(84.6%)	1.000
Sore throat ^f^	41	(77.4%)	13	(100.0%)	0.104
Lymadenopathy	21	(39.6%)	6	(46.2%)	0.909
Hepatosplenomegaly ^f^	4	(7.5%)	2	(15.4%)	0.337
Arthritis	23	(43.4%)	9	(69.2%)	0.174
Leukocytosis ^f^	44	(83.0%)	10	(76.9%)	0.691
Hepatitis ^f^	18	(34.0%)	2	(15.4%)	0.314
Systemic pattern ^f^	44	(83.0%)	4	(30.8%)	0.001 **
ESR(n = 46 vs. 12)	58.50	(31.00–107.25)	56.00	(33.5–84.75)	0.759
CRP(n = 46 vs. 12)	6.05	(1.80–14.50)	4.55	(2.43–14.70)	0.985
Ferritin(n = 46 vs. 12)	764.00	(298.50–3012.50)	1505.00	(462.00–3650.00)	0.520
Pouchot Score(n = 46 vs. 12)	5	(4–6)	5	(4–6)	0.526

Chi-Square test. ^f^ Fisher’s exact test. Mann-Whitney U test. * *p* < 0.05, ** *p* < 0.01.

**Table 4 life-11-00382-t004:** Risk factors for the occurrence of systemic pattern of disease outcome in patients with adult-onset Still’s disease.

	Univariate Analysis			Multivariate Analysis			
	OR	(95% CI)	*p* Value	OR	(95% CI)		*p* Value
Age	0.98	(0.94–1.02)	0.298				
Female	0.25	(0.05–1.22)	0.087				
Serum levels of IL-1β (pg/mL)	1.20	(0.89–1.62)	0.231				
Serum levels of IL-18 (pg/mL)	1.00	(1.00–1.00)	0.074				
Serum levels of CARD (pg/mL)	1.00	(1.00–1.00)	0.146				
Serum levels of Caspase-1 (pg/mL)	1.01	(1.003–1.026)	0.011 *	1.01	(1.002–1.025)		0.019 *
Genotype							
CT/TT	ref.			ref.			
CC	11.11	(2.78–50.00)	0.001 **	7.69	(1.67–33.33)		0.009 **
Rash	1.17	(0.27–5.13)	0.834				
Arthralgia	0.54	(0.11–2.79)	0.463				
Sore throat	0.48	(0.09–2.42)	0.370				
Lymphadenopathy	2.20	(0.68–7.14)	0.189				
Hepatosplenomegaly	1.98	(0.21–18.19)	0.547				
Arthritis	0.00	(0.00)	0.998				
Leukocytosis	0.87	(0.21–3.64)	0.845				
Hepatitis	1.75	(0.49–6.19)	0.385				

Logistic regression. * *p* < 0.05, ** *p* < 0.01.

## Data Availability

The data presented in this study are available in Appendix A.

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
