# Peer review of "CARD8 SNP rs11672725 Identified as a Potential Genetic Variant for Adult-Onset Still’s Disease"

_life, 2021, doi:10.3390/life11050382_

Round 1

Reviewer 1 Report

This study evaluated the associations of genetic polymorphisms of NLRP3-inflammasome signaling 24 with AOSD susceptibility and outcome and examine their functional property. I think these data are novel, and this manuscript is very well-written. I have only a question about haplotype about these genes. Did not the authors evalute haplotype study related with these genetic study?

Author Response

Thank you for the insightful question. We evaluate the CARD8 haplotype in 1000 Genomes phase 3 CHB (Han Chinese in Bejing, China) database, and there was no linkage data found. Please see the attachment.

Reviewer 2 Report

The manuscript describes that the SNP related to CARD8 is associated with the prevalence of AOSD and its systemic inflammation.

Although there are things that can be improved, the findings were novel. We find that the story is feasible and clear.

Major point

  1. The SNP analysis seen in table 1 has not undergone multiple comparison analyses. It is obvious that Bonferroni correction would not work in this case. We request the authors to use the false discovery rate and try to figure out that the significance is preserved. If not, please discuss this point in the discussion section.

Minor points

  1. Please clarify whether treatments were applied at the time the serum samples were obtained from the AOSD patients.
  2. IL-18 activity is known to be highly variable according to the abundance of IL-18 binding protein. The kit used didn’t indicate whether this is taken into account. Please clarify and discuss this point.
  3. In section 3.6, we didn’t see a clear outcome measure other than the systemic pattern. If possible please add the remission rate (Pouchot score etc.) and bio-DMARD induction rate stratified with the SNP.
  4. Serum ferritin and CRP are the biomarkers that are frequently used to assess inflammation in AOSD. If available adding these data may be helpful as they are more accessible in a clinical setting.

Author Response

Major point

  1. The SNP analysis seen in table 1 has not undergone multiple comparison analyses. It is obvious that Bonferroni correction would not work in this case. We request the authors to use the false discovery rate and try to figure out that the significance is preserved. If not, please discuss this point in the discussion section.

Thank for your good suggestion. We have adjusted the p-value with Benjamini & Hochberg correction among 53 candidate genes in the discovery set data. Neither p-value of CARS8 rs11672525 genotype or allele type were statistically significant (p=1.0 and p=0.477, respectively). Type I error was a concern. However, after taking the role of CARD8 in the pathogenesis of autoinflammatoy diseases into consideration, we still tried to sequence CARD8 rs11672525 directly in the validation dataset.

We discussed this point in section 4, on page 13, line 17-20, as follows:

We adjusted the p-value of 53 candidate genes in the discovery dataset with Benjamini & Hochberg correction. Neither p-value of CARS8 rs11672525 genotype or allele type were statistically significant (p=1.0 and p=0.477, respectively). Type I error was a concern”.

Minor points

  1. Please clarify whether treatments were applied at the time the serum samples were obtained from the AOSD patients.

Point 1: Thanks you for good comment. Serum samples from the AOSD patients were obtained after treatment. We add “We obtained serum samples from AOSD patients after treatment.” in section 2.1, page 3, line 1 for clarification.

We also added: ”We obtained serum samples after immunosuppressive treatment. We cannot exclude the potential interferences of immunosuppressants to inflammatory cytokines (IL-1 and IL-18) or inflammatory markers (CRP and ferritin). Future studies of treatment-naïve AOSD patients are needed to fully address this issue.” in section 4, page 13, lines 24-28.

  1. IL-18 activity is known to be highly variable according to the abundance of IL-18 binding protein. The kit used didn’t indicate whether this is taken into account. Please clarify and discuss this point.

Point 2: Thanks you for an excellent comment. We measure total IL-18 by a commercially available ELISA kit (Medical & Biology Laboratories Co, Ltd., Naka-ku, Nagoya, Japan), according to the manufacturer’s instructions. IL-18 binding protein were not measured. Imbalance of IL-18/IL-18BP in patients was reported in several diseases, such as hemophagocytic syndrome [Blood (2018); 131(13):1442–55] but IL-18BP were only moderately elevated even under the active inflammation stage. Sthoeger et. al also reported serum from SLE patients revealed high levels of both serum IL-18 and IL-18 binding protein. Despite the elevated levels of IL-18BP during active disease, free IL-18 remained 2-fold higher than the levels in healthy controls [J Autoimmun. (2010); 34(2):121-6]. Data above suggested IL-18BP protein may be reactive to serum IL-18 level, high disease activity, and may play a balancing role in inflammatory diseases.

We added the discussion of IL-18 BP in the discussion section (page 13, line 29-36) as follows: “In this study, we did no demonstrate the influence of IL-18 binding protein (IL-18 BP) on IL-18 activity and levels of free IL-18 of serum from AOSD patients. Imbalance of IL-18/IL-18BP in patients was reported in several diseases, such as hemophagocytic syndrome [38] and systemic lupus erythematosus [39], but IL-18BP was only moderately elevated even under the active stage. It suggested that IL-18BP protein may be reactive to serum IL-18 level, high disease activity, and may play a balancing role in inflammatory diseases. Further study to elucidate the relation between IL-18 BP and AOSD may be needed.

  1. In section 3.6, we didn’t see a clear outcome measure other than the systemic pattern. If possible please add the remission rate (Pouchot score etc.) and bio-DMARD induction rate stratified with the SNP.

Point 3: Thanks you for an insightful comment. However, there were no significant differences in Pouchot score between AOSD patients carrying CC genotype and CT/TT genotype (mean, [IQR]; 5, [4-6] vs 5, [4-6], p= 0.526).

We added the description of Pouchot score in Sec 2.1, page 3, lines 5-9: “Disease activity of each patient was assessed with Pouchot score, which assigned 1 point to each of 12 manifestations: fever, typical rash, pleuritis, pneumonia, pericarditis, hepatomegaly or abnormal liver function tests, splenomegaly, lymphadenopathy, leukocytosis > 15,000/mm3, sore throat, myalgia, and abdominal pain (maximum score: 12 points) [22] “ and modified table 3, page 10.

We had reviewed the literature, and the definition of Pouchot score remission was an absence of clinical symptoms and normalization of laboratory data. Since it was a cross-sectional study, we did not capture the remission rate longitudinally. Further cohort research focusing on remission and treatment is needed.

In this study, there were no bio-DMARD treated patients included. Pharmacogenomics study regarding bio-DMARD efficacy or side-effects awaits.

  1. Serum ferritin and CRP are the biomarkers that are frequently used to assess inflammation in AOSD. If available adding these data may be helpful as they are more accessible in a clinical setting.

Point 4: Thanks you for this useful suggestion. There were no significant differences of ESR, serum levels of CRP and ferritin between AOSD patients carrying CC genotype and CT/TT genotype (mean, [IQR]; 58.50, [31.00-107.25] vs 56.00, [33.5-84.75], p= 0.759; 6.05 [1.80-14.50] vs 4.55, [2.43-14.70], p=0.985; 764.00 [298.50-3012.50] vs 1505.00, [462.00-3650.00], p=0.520). We added data above in table 3, page 11.

We also modified Sec 3.5 in page 10, lines 14 as follows:” However, there was no significant difference in the proportion of the other manifestations or clinical data between AOSD patients carrying CC genotype and CT/TT genotype.”
